# Measurement report: The effects of SECA regulations on the atmospheric SO$_2$ concentrations in the Baltic Sea, based on long-term observations on the Finnish island, Utö

**Androniki Maragkidou**[1], **Tiia Grönholm**[1], **Laura Rautiainen**[2], **Juha Nikmo**[1], **Jukka-Pekka Jalkanen**[1], **Timo Mäkelä**[3], **Timo Anttila**[3], **Lauri Laakso**[2,4], **and Jaakko Kukkonen**[1,5]

[1]Atmospheric Composition Research, Finnish Meteorological Institute, Helsinki, Finland
[2]Marine Research, Finnish Meteorological Institute, Helsinki, Finland
[3]Climate System Research, Finnish Meteorological Institute, Helsinki, Finland
[4]Atmospheric Chemistry Research Group, Chemical Resource Beneficiation,
North-West University, Potchefstroom, South Africa
[5]Centre for Atmospheric and Climate Physics Research, University of Hertfordshire,
College Lane, Hatfield, UK

**Correspondence:** Androniki Maragkidou (androniki.maragkidou@fmi.fi)

**Abstract.** The designation of the Baltic Sea as a sulfur emission control area (SECA) in May 2006, with subsequent tightening of regulations in 2010 and 2015, has reduced the sulfuric emission from shipping traffic. This study assesses the impacts of SECA on observed SO$_2$ concentrations by providing a long-term analysis of 1 min time-resolution air quality data from 2006 to 2020 on Utö (an island in the Baltic Sea) supported by the predictions from the Ship Traffic Emissions Assessment Model (STEAM). Additionally, 1 h resolution data from 2003 to 2005 are utilized to investigate changes due to the SECA limits set in 2006. The observed SO$_2$ concentrations on Utö have continuously decreased since 2003 due to an overall decrease in SO$_2$ emissions in northern Europe, combined with reduced emissions from shipping traffic due to SECA regulations. The 3-year average SO$_2$ concentration dropped from pre-SECA (2003–2005) to post-SECA (2007–2009, 2011–2013, 2016–2018) periods by 38 %, 39 %, and 67 %, respectively. No clear trends were observed in the concentrations of other measured pollutants. Furthermore, we investigated wind-direction-resolved SO$_2$ concentrations for 2 selected years (2014 and 2019), and the results showed a significant decrease in high-SO$_2$-concentration shipping plumes due to the implementation of SECA in 2015. Our study brings out the importance of long-term, high-time-resolution air quality observations at a remote marine research station by providing means for both quantitative and qualitative analyses of the impacts of regulatory environmental legislation.

## 1 Introduction

In recent years, international maritime trade has experienced sustained growth, primarily due to its recognized economic efficiency, accounting for over 90 % of global trade (Patil et al., 2016). This growth has been projected to continue, with the United Nations Conference on Trade and Development (UNCTAD, 2019) indicating a prospective annual growth rate of 2.6 % to 3.4 % until 2024. In 2019, 11.08 billion tonnes of goods were shipped, and the global commercial fleet, consisting of 98 140 vessels exceeding 100 gross tonnes, had a total capacity of 2.06 billion deadweight tonnes (UNCTAD, 2021). In the Baltic Sea, the shipping activity increased fairly steadily during 2006–2020 (HELCOM, 2021).

For example, in 2020, the total number of fleet vessels increased by 271 %, 140 %, and 46 % in comparison to 2006, 2010, and 2015, respectively.

However, maritime activities have led to significant environmental challenges. In particular, shipping emissions are a major source of air pollution, contributing annually to approximately $20.88 \times 10^6$ t $NO_x$, $9.7 \times 10^6$ t $SO_2$, and $1.5 \times 10^6$ t $PM_{2.5}$ globally (Johansson et al., 2017). Smith et al. (2015) evaluated that global shipping is responsible for approximately 13 % and 12 % of total global anthropogenic emissions of $NO_x$ and $SO_x$, respectively. TS1 On a European scale, the European Environment Agency (EEA and EMSA, 2025) evaluated that 90 % of $SO_2$ emissions from transport was attributed to maritime transport, while $PM_{2.5}$, $PM_{10}$, and $NO_x$ transport-associated emissions accounted for 45 %, 28 %, and 35 %, respectively.

In 2013, in some areas of the Baltic Sea region, international shipping accounted for up to 80 % of the total concentrations of NO, $NO_2$, and $SO_2$ from all emission sources (Claremar et al., 2017), deteriorating the air quality in coastal areas (Jonson et al., 2015). Apart from their air quality impacts, these emissions also had the potential to cause acidification and eutrophication of marine waters and surrounding terrestrial ecosystems, with serious implications in the Baltic Sea environment (HELCOM, 2009; Hunter et al., 2011; Raudsepp et al., 2013).

Notably, the impacts of shipping emissions are not limited to sea areas. Around 70 % of ship emissions occur within 400 km of land (Corbett et al., 1999), while more than one-third of the world's population lives within 100 km from the coast (UNEP, 2024). Pollution from shipping can reach the inhabited land areas and have a severe impact on human health (Corbett et al., 1999, 2007; Eyring et al., 2010; Matthias et al., 2010; Tian et al., 2013; Anenberg et al., 2019). Shipping emissions have been shown to negatively impact the air quality, especially in coastal regions and portside (Chen et al., 2019; Donateo et al., 2014; Liu et al., 2017); the climate (Contini et al., 2015; Merico et al., 2016; Sofiev et al., 2018); and the economy (Jalkanen et al., 2014). Andersson et al. (2009) and Brandt et al. (2013) evaluated that international shipping was responsible for approximately 50 000 premature deaths annually in Europe. Furthermore, Barregard et al. (2019) reported that shipping in the Baltic Sea and the North Sea alone may have resulted in approximately 14 000 premature deaths in Europe in 2011.

While sulfur reductions have improved air quality and subsequently human health (Jonson et al., 2015; Sofiev et al., 2018; Barregard et al., 2019), it should be kept in mind that reducing sulfuric emissions from shipping may lead to unintended adverse impacts on climate and marine ecosystems. The impact on climate, due to a decrease in the cooling effect of ship-emitted aerosols, is likely leading to an increase in global sea surface and atmospheric temperatures (Lauer et al., 2009; Partanen et al., 2013; Hausfather and Forster, 2023). The impacts of sulfur reduction on marine ecosys-

tems includes concentrated aquatic-phase pollutant emissions from the exhaust gas cleaning systems, as discussed in Picone et al. (2023) and Hermansson et al. (2024). Thus, the potential success of sulfur reduction regulations discussed in this study may require further investigations in a wider context not focusing only on direct human health benefits.

As international maritime trade and shipping emissions are increasing, measures to limit and monitor their adverse impacts are necessary. To mitigate shipping emissions, the International Maritime Organization (IMO) has implemented a series of regulations aimed at reducing emissions from ships. These regulations include the establishment of sulfur emission control areas (SECAs), which require ships to use low-sulfur fuel in designated areas or, alternatively, reduce the sulfuric emission to air with an exhaust gas cleaning system. In May 2006, the Baltic Sea was designated as a SECA along with the North Sea and the English Channel. SECAs have also been set up along the North American east and west coast and the US Caribbean. The maximum allowed sulfur content of fuel in the SECAs was decreased in 2006 from 2.7 % to 1.5 %. On 1 July 2010, the International Maritime Organization (IMO) implemented a rule that required ships sailing in SECAs to use fuel with a sulfur content no higher than 1.0 % (Van Aardenne et al., 2013). This rule was further tightened in 2015, reducing the sulfur limit to 0.1 %.

Given these regulations, it is essential to assess their effectiveness through compliance studies across various regions. For instance, Beecken et al. (2015) conducted two campaigns in the Gulf of Finland and Neva Bay during the summers of 2011 and 2012, sampling 466 plumes from 311 individual vessels using both ground-based and helicopter-borne measurements. Their analysis revealed that 90 % of the observed plumes complied with the 1 % sulfur content limit set for SECAs in 2011, with this compliance increasing to 97 % in 2012. Similarly, Yang et al. (2016) observed a significant decrease in ship-emitted $SO_2$ following the implementation of SECA regulations in January 2015 in the English Channel, underscoring the substantial impact of these policies on air quality. Mellqvist et al. (2017) reported compliance levels of 92 %–94 % around Denmark during 2015–2016, and Jonson et al. (2019) revealed strong compliance with SECA regulations in the Baltic Sea based on emission modelling. Additionally, Repka et al. (2019) conducted a comprehensive study and found compliance rates exceeding 98 % in Göteborg and Gdynia, 95 %–97 % in St. Petersburg, and 94 % in the central Baltic Sea. However, compliance rates were slightly lower at SECA borders, around 87 %.

In this study, the impacts of SECA regulations on $SO_2$ concentrations on a small island in the Baltic Sea are studied. More specifically, this research will present unique fine-temporal-resolution data of $SO_2$, $NO_x$, NO, $PM_{2.5}$, and $O_3$ concentrations measured on the island Utö at the outer edge of the Archipelago Sea in the Baltic Sea, providing further insights into the impacts and the effectiveness of SECA regulations. The observations were carried out during the period

2006–2020, covering a period with three separate changes in SECA limits. Additionally, sparser hourly data from 2003 to 2005 were also utilized.

5 Moreover, the observations of air quality parameters are supported with local meteorological data and emission estimates utilizing AIS data. To further contextualize these measurements, we also computed the emissions attributed to shipping in the Baltic Sea during the target period using the Ship Traffic Emissions Assessment Model (STEAM). It is 10 also noteworthy that the dataset used in this study, which includes wind direction analysis of $SO_2$ concentrations relative to a major shipping lane, has not previously been published or analysed.

## 2 Measurement location and site characteristics

15 The observations utilized in this study were carried out on Utö ($59°46'50$ N, $21°22'23$ E), a small island in the Baltic Sea (Fig. 1a) with an area less than $1\,km^2$. It is located $70\,km$ from the coast of the mainland of Finland and surrounded only by the open sea and a few smaller islands. The vegeta- 20 tion between the bare rocky areas consists mainly of different shrubs and a few trees. The island is flat, and the highest point is less than $15\,m$ above the sea surface. Fewer than 40 inhabitants live on Utö during the winter. The population, including tourists during the summer months, varies between 100 25 and 200. Ships that pass Utö are pilot, passenger, military, fishing, cargo, commercial, and tanker vessels. An important shipping lane to or from Finland is located at a distance of approximately $1\,km$ west of the island (Fig. 1b).

Meteorological observations on Utö started in 1881. Since 30 the beginning of observations, there has been a gradual increase in measured atmospheric, air quality, marine, and electromagnetic parameters (Ahlnäs, 1961; Laurila and Hakola, 1996; Komppula et al., 2007; Kyllönen et al., 2009; Kilkki et al., 2015; Laapas and Venäläinen, 2017; Laakso et 35 al. 2018; Honkanen et al., 2018, 2021, 2024; Grönholm et al., 2021; Kraft et al., 2021; Seppälä et al., 2021; Rautiainen et al., 2023; Hellén et al., 2024). As the island is sparsely populated and there is very little land traffic, there are no local sources of $NO_x$ and $SO_2$. Consequently, Utö is an ideal 40 location for plume tracking.

The long-term marine and meteorological observations and other characteristics of Utö Atmospheric and Marine Research Station are discussed in Laakso et al. (2018) and the references listed above; thus only few results relevant 45 for this study are summarized here. The prevailing meteorological conditions in the area are characterized by high wind speeds, with monthly averages ranging from $5.6\,m\,s^{-1}$ in July to $8.9\,m\,s^{-1}$ in December, resulting in an annual average of $7.1\,m\,s^{-1}$. The prevailing wind is from the southwest, 50 while the sector from south (clockwise) to northwest dominates, accounting for approximately 60 % of the observed wind patterns (Honkanen et al., 2018). An examination of the

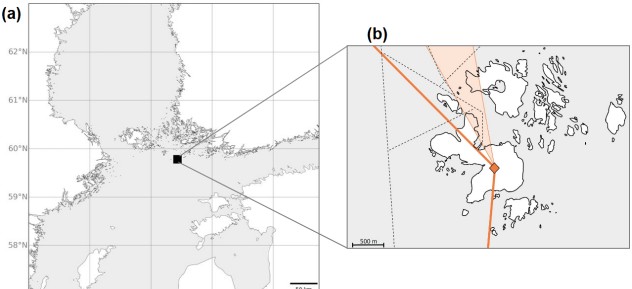

**Figure 1. (a)** The location of Utö in the Baltic Sea and **(b)** the Utö air quality station (orange diamond) and the shipping lanes adjacent to the island (dashed black lines). Major ships operate on the north–south shipping lane at a distance of approximately $500\,m$ west of Utö (or $1\,km$ from the air quality station). The wind sector representing the directions from the main shipping lane ($185–315°$) is indicated by the orange lines. The ship traffic to and from the island's harbour is indicated by the shaded orange sector.

wind time series from 1959 to 2016 (Laapas and Venäläinen, 2017) indicated no substantial alterations in wind direction or wind speed, aligning with a recent investigation covering 55 the period of 1979 to 2008.

## 3 Measurements and methods

### 3.1 Air quality and wind observations

To study the effects of SECA regulations on air quality, concentrations of $SO_2$, $PM_{2.5}$, NO, $NO_x$, and $O_3$ were mea- 60 sured at 1 min time resolution and analysed. The study period spanned 15 years, from 2006 to 2020 (supporting hourly data from 2003 to 2005). During this time, some of the instruments were changed (Table 1). The data went through normal quality control procedures as defined by European Monitor- 65 ing and Evaluation Programme (EMEP) and World Meteorological Organization (WMO) standards (see e.g. Anttila and Tuovinen, 2010, and references therein). In addition to the air quality observations, wind data on site were used. The temporal resolution of the wind data was 10 min; therefore it 70 was interpolated linearly for intervals of 1 min to match the air quality data.

Calibration of the instruments is carried out with precision to ensure accuracy and reliability. Initially, instruments are calibrated at the National Reference Laboratory for Air 75 Quality at the Finnish Meteorological Institute (FMI). Following this, instruments are calibrated in the field using reference values from the laboratory, involving zero level and three span points to confirm linearity.

The $SO_2$ instrument is calibrated with the Gasmet Ansyco 80 system, which utilizes a permeation chamber to produce $SO_2$ test gas and can also generate zero air. For $PM_{2.5/10}$, the Fidas instrument is calibrated according to the manufacturer's procedure using a MonoDust (reference particles) provided

**Table 1.** Air quality observations.

| Observation | Device | Start | End | Notes | Uncertainty* |
|---|---|---|---|---|---|
| $SO_2$ | Thermo 43i-TLE | 9 Sep 2011 | | | 1 % of reading or 1 ppb (whichever is greater) |
| | Thermo 43s | 1 Jan 1996 | 8 Sep 2011 | | 1 % of reading or 1 ppb (whichever is greater) |
| $PM_{2.5}$ | Fidas 200E 5030 Sharp | 11 Jun 2021 22 Nov 2017 | | | 9.7 % $\pm 2.0\,\mu g\,m^{-3} < 80\,\mu g\,m^{-3}$; $\pm 5\,\mu g\,m^{-3} > 80\,\mu g\,m^{-3}$ (24 h) |
| | Thermo FH62 I-R | 1 Jun 2003 | 22 Nov 2017 | | $\sim 17\,\%**$ |
| $NO_x$ | Thermo 42i-TL | 7 Feb 2014 | | | $\pm 0.4$ ppb in the range of 0–500 ppb |
| | Horiba APNA-370 | 18 Aug 2007 | 7 Feb 2014 | | 0.5 ppb |
| | TEI 42CTL | | 18 Aug 2007 | Data gap 2006–2007 | $\pm 0.4$ ppb in the range of 0–500 ppb |
| $O_3$ | Thermo 49i | 12 Feb 2015 | | | 1 ppb |
| | Horiba Ltd APOA-360 | 3 Oct 2003 | 11 Feb 2015 | | $\pm 1.0\,\%$ |

* Based on instruments' or equivalent models' data sheets, manuals, or manufacturer-provided specifications available online. ** Based on Lagler et al. (2011).

with the instrument to define the number of particles entering the device. Zero calibration is achieved with a HEPA filter, eliminating the need for reference values from the National Reference Laboratory.

$NO_x$ is calibrated using the Sonimix 3012/3022 multigas calibrator, which includes a gas-phase titration (GPT), ozone, and a self-regenerating zero air generator. Calibration for NO specifically is conducted with externally supplied diluted NO in $N_2$. $O_3$ is calibrated with the 49i-PS, a UV photometric primary standard designed for the calibration of ozone analysers. These procedures ensure that measurements are both accurate and reliable.

## 3.2 Automatic identification system (AIS) data

The automatic identification system (AIS) is a mechanism for the automated generation and transmission of vessel-related information to both other vessels and coastal authorities (IMO, 2020). The AIS data encompassed main ship attributes, including Maritime Mobile Service Identity (MMSI), latitude, longitude, true heading, course over ground, and speed over ground. Based on the HELCOM AIS data, vessels were categorized into six distinct types that best characterized their specific roles: cargo ships, large passenger ships, medium-sized passenger vessels, large work vessels, small vessels, and others. In the vicinity of Utö, cargo and large passenger ships constituted the majority of vessels.

In our study, AIS data were utilized to calculate emissions using STEAM and to investigate potential changes in the shipping traffic density close to the Utö measurement site.

## 3.3 STEAM

STEAM integrates AIS-derived data alongside technical insights into the global fleet and fundamental naval architecture principles (Jalkanen et al., 2009, 2012, 2014; Johansson et al., 2017). The model is subsequently used to predict vessel water resistance and instantaneous engine power of the main and auxiliary engines. STEAM facilitates the prediction of instantaneous fuel consumption and the emissions of specific pollutants (Jalkanen et al., 2016).

The input data for STEAM concerning ship properties encompasses, among other factors, measured emission factors (when available), shaft generators, specific fuel oil consumption, and fuel type and sulfur content for main and auxiliary engines, along with installed abatement techniques (Jalkanen et al., 2009). The computed emissions for $NO_x$ used in this study encompass the International Maritime Organization (IMO) registered traffic. The assessment includes all vessel traffic equipped with AIS transceivers while specifically excluding those navigating in inland waterways from the dataset. In this work, STEAM was used to run the Baltic Sea ship emission time series between the years 2006 and 2020, including the regulatory changes to sulfur content of

marine fuels. Furthermore, STEAM was used to study the vessel traffic statistics near Utö and determine hourly vessel counts. STEAM was also used as a supporting tool in the interpretation of air quality measurement results to determine the emission intensity of local plumes.

## 4 Results and discussion

The results are divided into subsections addressing the changes in long-range transport and shipping, investigating (i) the air quality time series and (ii) the wind-direction dependence of observations. For convenience, we will use the abbreviations SECA2006, SECA2010, and SECA2015 throughout the text to refer to the SECA regulations established on 19 May 2006, 1 July 2010, and 1 January 2015, respectively.

### 4.1 Changes in long-range transport and shipping emissions

The $SO_2$ emissions in northern Europe have shown a decreasing trend in most of the emission sectors. Figure 2 illustrates the combined emissions in each emission sector from countries that have a major impact on the northern Baltic Proper/southern border of Archipelago Sea during the period 2003–2020. The dominant emissions originate from the energy (indicated as "public power" in Fig. 2) and industry sectors, exceeding the emissions from shipping by more than 1 order of magnitude. The average lifetime of $SO_2$ in the lower atmosphere is approximately 1–3 d (Lee et al., 2011; Beirle et al., 2014). Part of the $SO_2$ observed on Utö therefore originates from long-range transport of regional background pollution, while another portion is attributed to shipping traffic in the vicinity of Utö (e.g. Lee et al., 2011). Long-range transported background pollution is well diluted, whereas local emissions tend to manifest as short-term, high-concentration peaks.

Figure 3 displays the monthly shipping emissions in the Baltic Sea calculated using STEAM. The changes in $SO_2$ emissions are a combination of changes in emissions due to the implementation of SECA regulations and changes in shipping traffic density.

From 2006 onward, shipping activity in the Baltic Sea has generally increased, with some exceptions. In particular, notable downturns in the total number of ships and the transport work were observed during the economic recession, which started in 2008 and continued until 2009, affecting most areas of the neighbouring countries of the Baltic Sea (Jalkanen et al., 2014), and again during the COVID-19 pandemic in 2019 (HELCOM, 2021). An exception to this upward trend occurred in 2013, with approximately 350 000 ship crossings in the used passage lines compared to 2012, likely due to the reductions in cargo ships attributed to reduced shipping activity resulting from the economic recession during these years (HELCOM, 2014). Finland also experienced a unique gross

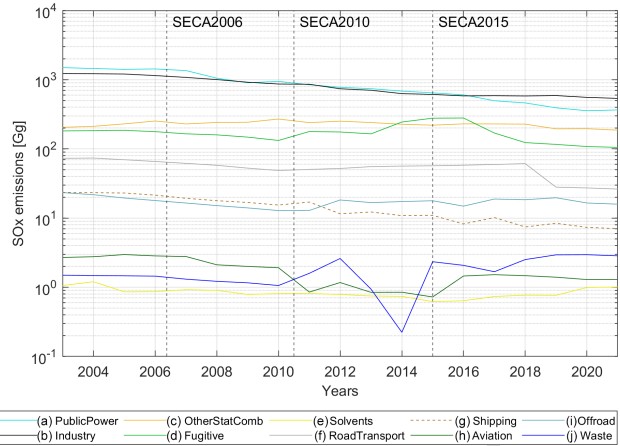

**Figure 2.** The combined annual $SO_x$ emissions of Estonia, Finland, Lithuania, Latvia, Sweden, Russia, and Poland divided into source sectors during the period 2003–2020. The implementation of subsequent SECA regulations is indicated with dashed vertical lines. Note that the figure is presented on a logarithmic scale ($y$ axis). The source of $SO_x$ emissions data for the different sectors is the European Monitoring and Evaluation Programme (EMEP).

domestic product (GDP) decline from 2011 to 2012, largely due to weak export performance (Bank of Finland, 2018).

Emission computations from STEAM (Fig. 3) indicate a gradual decline in $SO_x$ and $PM_{2.5}$ emissions attributed to maritime traffic in the Baltic Sea starting in 2006. This decline in $SO_x$ and $PM_{2.5}$ emissions continued gradually until 2014, at which point Baltic economies began to recover, reaching pre-crisis production levels (Bank of Finland, 2018). Following the introduction of SECA2015, a sharp decrease in $SO_x$ and $PM_{2.5}$ emissions is observed, highlighting the impact of stricter emissions controls. The magnitude of the relative reduction in $PM_{2.5}$ emissions attributed to shipping is smaller than that of the $SO_x$ emissions. The predicted annual $NO_x$ emissions during the period between 2006 and 2020 have remained relatively stable, except for between 6 November and 31 December 2017, when a 95 % decrease in AIS data was observed due to an issue with AIS data reception.

### 4.2 Observed changes in ship traffic and concentrations of $SO_2$ and other air quality parameters on Utö

On Utö, local marine traffic typically consists of a few larger vessels per day. We analysed the AIS data to quantify the number of ships passing within 5 km of the island of Utö on a daily and yearly basis (Fig. 4). As expected, results showed a declining trend in the yearly number of ships up to 2010, which coincided with the 2008–2009 economic recession affecting the majority of the Baltic Sea region. The decline continued through 2013, with a slight increase observed in 2014, as production levels across the region began to recover (Bank of Finland, 2018).

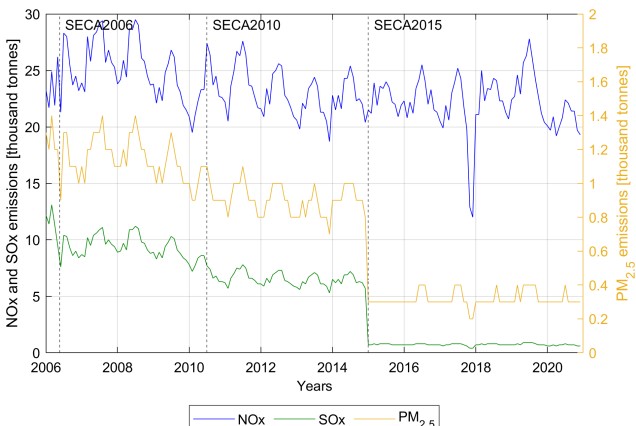

**Figure 3.** Predicted emissions of $NO_x$, $SO_x$, and $PM_{2.5}$ attributed to maritime traffic in the Baltic Sea from January 2006 to December 2020, computed using STEAM. The SECA regulations introduced in 2006, 2010, and 2015 are represented by dashed vertical lines.

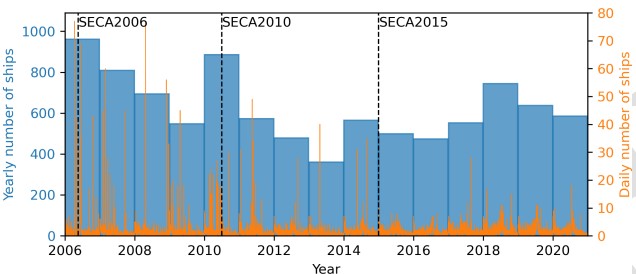

**Figure 4.** The total number of ships per year (left $y$ axis) that have passed the island of Utö at distances closer than 5 km (vertical blue bars). The total number of ships per day (orange line) is shown on the right $y$ axis.

From 2015 to 2020, however, no clear increasing or decreasing trend in the number of ships was detected.

The observed 30 d moving averages and moving percentiles of the concentration data for $SO_2$, $PM_{2.5}$, NO, $NO_x$, and $O_3$ from 2006 to the end of 2020 are presented in Fig. 5. The dashed vertical lines represent the introduction of SECA2006, SECA2010, and SECA2015, corresponding to 1.5 %, 1.0 %, and 0.1 % sulfur fuel content limits, respectively. The effects of the sulfur cap in 2006 and 2015 are clearly visible, while the impact of 2010 is negligible. The change in 2015 is apparent in the shifts in the 1st and 99th percentiles corresponding to short-duration local ship plumes. Additionally, the mean and median values after 2015 are closer to each other, indicating that the higher peaks in the data occur less frequently (Fig. 5a).

In general, the reduction in the release of $SO_2$ emissions attributed to shipping could also result in a decline in the production of particulate matter originating from precursor compounds from shipping. However, as the results (Fig. 5 and Appendix A) demonstrate, there were no step changes in the

**Table 2.** Annual average and median concentrations of $SO_2$ on Utö. Values for the period 2003–2005 are based on hourly averages, while, for the period 2006–2020, 1 min time-resolution data are used. STD is the standard deviation. $N$ (%) represents the fraction of the year for which high-quality data are available, expressed as a percentage. High-quality data, used in our analysis, are defined as valid data recorded, excluding those compromised by factors such as instrument malfunctions, environmental interference, or calibration issues.

| $SO_2$ | Mean ($\mu g\,m^{-3}$) | Median ($\mu g\,m^{-3}$) | STD ($\mu g\,m^{-3}$) | $N$ (%) |
|---|---|---|---|---|
| 2003 | 1.38 | – | 2.15 | 98.3 |
| 2004 | 1.17 | – | 1.13 | 91.5 |
| 2005 | 1.32 | – | 1.45 | 97.1 |
| 2006 | 0.93 | 0.55 | 1.51 | 55.7 |
| 2007 | 0.97 | 0.56 | 1.58 | 99.8 |
| 2008 | 0.70 | 0.40 | 1.19 | 98.9 |
| 2009 | 0.73 | 0.42 | 1.07 | 98.4 |
| 2010 | 1.12 | 0.58 | 1.80 | 89.0 |
| 2011 | 1.00 | 0.57 | 1.81 | 93.0 |
| 2012 | 0.78 | 0.45 | 1.18 | 95.8 |
| 2013 | 0.58 | 0.38 | 0.81 | 84.5 |
| 2014 | 0.75 | 0.46 | 1.08 | 99.5 |
| 2015 | 0.37 | 0.29 | 0.47 | 99.4 |
| 2016 | 0.45 | 0.32 | 0.68 | 99.4 |
| 2017 | 0.34 | 0.27 | 0.54 | 98.3 |
| 2018 | 0.50 | 0.36 | 0.60 | 99.3 |
| 2019 | 0.41 | 0.36 | 0.50 | 99.1 |
| 2020 | 0.34 | 0.32 | 0.35 | 99.1 |

$PM_{2.5}$ concentration during the considered period. This does not mean that the emissions of particulate matter from ships were at the same level as before the introduction of SECA regulations but rather that the measurement of $PM_{2.5}$ is not sensitive enough to show the changes in the ultrafine particle size range. $PM_{2.5}$ concentrations also had some negative values, which were removed, leading to gaps in the time series (Fig. 5b). Seppälä et al. (2021) studied the impact of SECA on particle number size distribution (size range 7–537 nm) on Utö and found that the main changes due to SECA in particle number occurred in sizes between 33 and 144 nm. Thus, the impact on particle mass, resulting mainly from larger particles, is minor. Similarly to $PM_{2.5}$, no decreases were observed in the $NO_x$, NO, and $O_3$ concentrations. However, for NO and $NO_x$, a period of data from 22 May 2010 to 15 June 2011 was removed (Fig. 5c and d) due to abnormally low values, likely caused by overly strict data processing. The year-to-year temporal variation in $PM_{2.5}$, NO, $NO_x$, and $O_3$ (yearly mean and median values and standard deviation) is presented in Appendix A.

Since the years 2006, 2010, and 2015 marked significant milestones, as they represented the introduction of the new SECA regulations, we decided to focus on the periods that occurred before and after the implementation of SE-

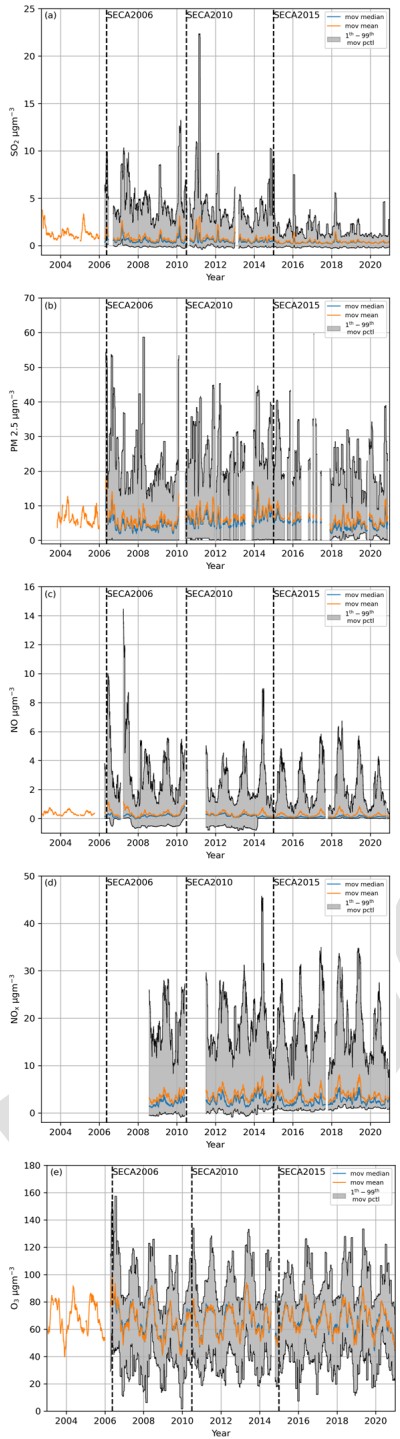

**Figure 5.** Time series of measured concentrations for **(a)** $SO_2$, **(b)** $PM_{2.5}$, **(c)** NO, **(d)** $NO_x$, and **(e)** $O_3$ from 2003 to the end of 2020 at the air quality station on Utö. The concentrations are presented as moving percentiles based on 1 min resolution data within a 30 d window, with 30 d moving mean, median, and 1st–99th percentiles. For the years 2003–2005, only the moving mean is shown, based on 1 h resolution data. The implementation of SECA regulations in 2006, 2010, and 2015 is indicated by three dashed vertical lines.

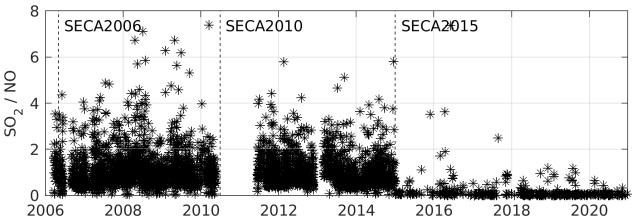

**Figure 6.** Time series of the observed $SO_2$/NO concentration ratio, when the wind was blowing from the westerly direction (wind direction 180–360°) and the plume from the passing ships was transported over Utö. Only the periods with simultaneous peaks in both $SO_2$ and NO signals were included into the analysis.

CAs: 2003–2005, 2007–2009, 2011–2013, and 2016–2018. Specifically, we compared the 3-year average of $SO_2$ concentrations before the SECA introduction (2003–2005) to those following each SECA enactment (2007–2009, 2011–2013, and 2016–2018) (Table 2). Our findings revealed that 3-year average $SO_2$ concentrations from the pre-SECA period (2003–2005) decreased by 38 %, 39 %, and 67 % in comparison to the post-SECA periods (2007–2009, 2011–2013, and 2016–2018), respectively.

As indicated in the previous section, $SO_2$ emissions have been widely reduced across various industrial sectors. To estimate the local effect of shipping, we utilized NO, which is highly reactive and has a short lifetime in the air. Since it is one of the main gases in ship exhaust, it can serve as a marker for ship plumes originating near Utö. Therefore, we selected periods when $SO_2$ and NO both showed clear peaks simultaneously in the data and when the wind was blowing from the western sector (180–360°). The prominence of the peak indicating the peak height relative to other data was chosen to be $2 \, \mu g \, m^{-3}$ for both NO and $SO_2$. Figure 6 depicts the $SO_2$ concentration normalized by NO concentration during these plumes. Normalization using $CO_2$ concentrations would have allowed further analysis of fuel sulfur content; unfortunately, such data were not measured on Utö in a location suitable for ship $SO_2$ plume research prior to the implementation of SECA in 2015. There is a clear difference before and after January 2015. Prior to the implementation of the strictest SECA regulation (i.e. SECA2015), the $SO_2$ concentrations during local pollution plumes were of the same magnitude as NO. However, after the SECA2015 regulation was enacted, the $SO_2$ concentrations decreased significantly within these near-source plumes.

## 4.3 Dependence of concentrations on local wind direction

To study the impacts of ships passing by Utö, we selected $SO_2$ concentration data based on wind direction. Firstly, we separated the data points measured when the wind was blowing from the shipping lane (covering wind directions from

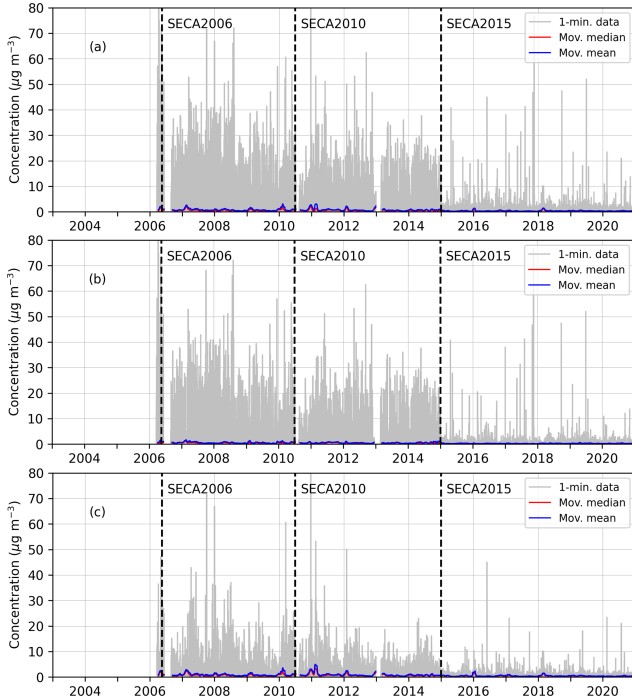

**Figure 7.** The time series of measured $SO_2$ concentrations (1 min temporal resolution; solid grey lines) and their moving means (solid blue lines) and medians (solid red lines) in a 30 d window with **(a)** all the data points included (unstratified); **(b)** data only from a 130°-wide wind sector from the direction of shipping lane, extending from 185 to 315°; and **(c)** data only from background wind sectors excluding the shipping lane, less than 185° and over 315°. The temporal resolution of the $SO_2$ concentration data was 1 min on average.

185 to 315°) towards the measurement site from the rest of the data.

All data, along with data from when the wind was blowing from the direction of the shipping lane and data from ₅ the background sector (wind directions excluding the shipping lane sector), are shown in Fig. 7a–c. Similarly to the previously presented results, there is an evident decrease in $SO_2$ concentrations after SECA2015 and a slight decrease after SECA2010. This decreasing trend is visible in all three ₁₀ plots, but the most pronounced decrease occurs after 2015, when only wind directions from the shipping lane were considered.

To analyse the effect of wind direction in more detail, we generated concentration rose plots in finer 10° intervals ₁₅ (Fig. 8). To assess the effects of the implementation of SECA regulation in 2015, we chose 2 years: one immediately preceding the new regulation, 2014, and the other representing the 5-year period after, 2019. The years 2016, 2017, 2018, and 2019 were not substantially different in terms of $SO_x$ ₂₀ emissions from shipping (see Fig. 2) or the number of ships (Fig. 4). In this regard, any of these years could have been selected as an example year for the post-SECA 2015 analysis.

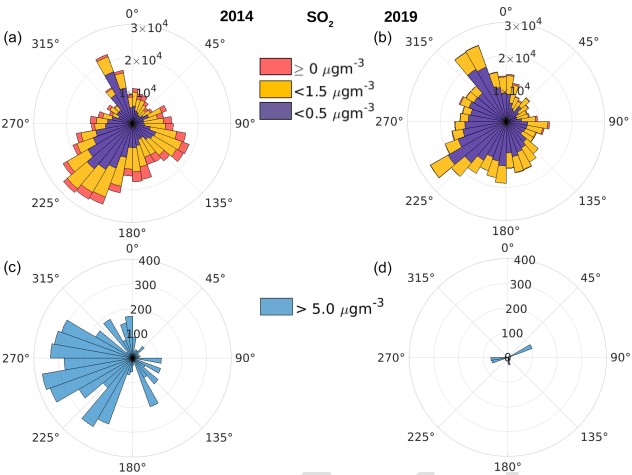

**Figure 8.** The measured concentrations of $SO_2$ on the island of Utö as a function of the wind direction, as polar histograms with 1 min resolution. Panels **(a)** and **(c)** correspond to the data in 2014, and panels **(b)** and **(d)** correspond to the data in 2019. These 2 years were selected to represent the situation before and after the SECA regulation in 2015. The upper panels **(a)** and **(b)** include all data points, whereas the lower panels **(c)** and **(d)** include only the data for which the concentrations were higher than a selected threshold value ($5 \mu g\,m^{-3}$). In panels **(a)** and **(b)**, the red in the legend represents all data points with values larger than $0 \mu g\,m^{-3}$. Overlaid on the red, the yellow shows data points with values below $1.5 \mu g\,m^{-3}$, and the violet, which overlays the yellow, indicates data points with values below $0.5 \mu g\,m^{-3}$. The radial axis represents the number of measured cases for each wind direction sector.

The selected year 2019 was prior to the COVID-19 pandemic and selected for comparison. The pandemic did not affect the emissions in Europe in 2019; its effects were felt only during ₂₅ the subsequent years.

The measured annual distribution of wind direction is determined by the local climate at the site and potentially by local factors affecting wind measurements. The most common wind directions in this region are southwesterly, with rela- ₃₀ tively high contributions also from northwesterly directions at angles from 320 to 350° (Fig. 8a and c). This northwesterly sector coincidentally corresponds to the routes of shipping to and from the harbour and wharf area of the island. The shipping traffic to the island typically consists of smaller ₃₅ vessels on average compared to the average marine transport in the neighbouring ship lane. The upper panels in Fig. 8a–b include all measured concentrations. The observed $SO_2$ concentrations were systematically and substantially higher in 2014 compared to the values in 2019. Furthermore, if only ₄₀ peak concentrations during the plumes are considered (lower panels, Fig. 8c–d), it is evident that the numbers of the highest concentrations were drastically higher in 2014 compared to the corresponding values in 2019 and originated from the direction of the adjacent shipping lane (i.e. from the westerly ₄₅ direction).

## 5 Conclusions

In this study, we presented and analysed a high-temporal-resolution air quality dataset of $SO_2$, $PM_{2.5}$, NO, $NO_x$, and $O_3$ concentrations and of wind data for 1.5 decades on the island of Utö in the Baltic Sea. This dataset is particularly unique, as there are no other similar long-term air quality observations from remote locations in the Baltic Sea, and it has not previously been published or analysed. Utö's central location next to northern Baltic Proper, with minimal influence from other pollution sources, makes it an ideal site to study long-term pollution trends from shipping. The period of the study includes all three consecutive revisions of the SECA regulations.

The findings derived from the analysis of the air quality and meteorological datasets and STEAM indicated that the SECA regulations, introduced in 2006, tightened in 2010, and further strengthened in 2015, have been successful in reducing the observed $SO_2$ concentrations on Utö during the studied period 2003–2020. Comparing 3-year average $SO_2$ concentrations pre-SECA (2003–2005) to post-SECA periods (2007–2009, 2011–2013, 2016–2018) revealed reductions of 38 % (SECA 2006), 39 % (SECA 2010), and 67 % (SECA 2015), respectively. During the target period, there were no clear decreasing trends for the concentrations of $PM_{2.5}$, $NO_x$, NO, and $O_3$. The year-to-year variations in the concentrations were substantial for all pollutants; these were attributed partly to the variations in regional meteorology and partly to the variations in emissions.

We also analysed polar histograms of $SO_2$ concentration during 2 representative years, 2014 and 2019, i.e. before and after the implementation of the SECA2015. This analysis was done both for the whole dataset and for selecting only the highest measured concentrations above a specified threshold. The number of cases with the highest $SO_2$ concentrations was drastically higher in 2014 compared to the corresponding values in 2019. The highest concentrations mostly originated from the direction of the adjacent shipping lane. This analysis therefore indicates that the highest measured concentrations attributed to shipping have substantially decreased after SECA2015.

By filling an important gap in the current literature, this study provides a thorough view of the air quality trends in the Baltic Sea and highlights the importance of good-quality, high-temporal-resolution, long-term air quality data at remote marine research stations. Such observations are crucial for both quantitative and qualitative analyses of the impacts of regulatory environmental legislation. The findings of this work will provide valuable insights into the effectiveness of SECA regulations and serve as a benchmark for local and regional dispersion modelling efforts for future research utilizing the data of this study.

## Appendix A

**Table A1.** Annual average and median concentrations of $PM_{2.5}$ on Utö. Values for the period 2003–2005 are based on hourly averages, while, for the period 2006–2020, 1 min time-resolution data are used. STD is the standard deviation. $N$ (%) represents the fraction of the year for which high-quality data are available, expressed as a percentage. High-quality data, used in our analysis, are defined as valid data recorded, excluding those compromised by factors such as instrument malfunctions, environmental interference, or calibration issues.

| $PM_{2.5}$ | Mean ($\mu g\,m^{-3}$) | Median ($\mu g\,m^{-3}$) | STD ($\mu g\,m^{-3}$) | $N$ (%) |
|---|---|---|---|---|
| 2003 | 5.97 | – | 5.57 | 20.0 |
| 2004 | 6.82 | – | 6.07 | 84.8 |
| 2005 | 6.07 | – | 6.13 | 93.8 |
| 2006 | 8.53 | 5.67 | 9.44 | 68.9 |
| 2007 | 5.28 | 3.93 | 5.49 | 99.9 |
| 2008 | 5.25 | 3.87 | 5.99 | 98.8 |
| 2009 | 5.30 | 4.37 | 4.84 | 98.0 |
| 2010 | 7.73 | 6.02 | 7.07 | 58.2 |
| 2011 | 7.60 | 6.02 | 7.23 | 79.3 |
| 2012 | 6.14 | 4.70 | 5.54 | 72.0 |
| 2013 | 5.92 | 4.54 | 5.32 | 58.5 |
| 2014 | 8.97 | 7.24 | 7.34 | 82.4 |
| 2015 | 7.06 | 5.66 | 6.15 | 71.3 |
| 2016 | 6.38 | 5.28 | 5.15 | 62.1 |
| 2017 | 5.78 | 4.27 | 6.31 | 63.1 |
| 2018 | 5.61 | 3.85 | 4.93 | 97.3 |
| 2019 | 5.56 | 4.14 | 4.53 | 89.5 |
| 2020 | 5.51 | 3.99 | 5.00 | 86.2 |

**Table A2.** Annual average and median concentrations of NO on Utö. Values for the period 2003–2005 are based on hourly averages, while, for the period 2006–2020, 1 min time-resolution data are used. STD is the standard deviation. $N$ (%) represents the fraction of the year for which high-quality data are available, expressed as a percentage. High-quality data, used in our analysis, are defined as valid data recorded, excluding those compromised by factors such as instrument malfunctions, environmental interference, or calibration issues.

| NO | Mean ($\mu g\,m^{-3}$) | Median ($\mu g\,m^{-3}$) | STD ($\mu g\,m^{-3}$) | $N$ (%) |
|---|---|---|---|---|
| 2003 | 0.40 | – | 0.76 | 98.3 |
| 2004 | 0.39 | – | 0.60 | 91.1 |
| 2005 | 0.35 | – | 0.57 | 70.4 |
| 2006 | 0.46 | 0.14 | 1.85 | 71.3 |
| 2007 | 0.40 | 0.16 | 1.62 | 86.9 |
| 2008 | 0.29 | 0.15 | 1.30 | 99.1 |
| 2009 | 0.33 | 0.22 | 1.04 | 97.9 |
| 2010 | 0.64 | 0.48 | 1.27 | 35.9* |
| 2011 | 0.33 | 0.25 | 0.98 | 50.2* |
| 2012 | 0.33 | 0.24 | 1.11 | 98.1 |
| 2013 | 0.30 | 0.21 | 1.14 | 98.0 |
| 2014 | 0.29 | 0.09 | 1.09 | 99.4 |
| 2015 | 0.22 | 0.08 | 1.08 | 98.6 |
| 2016 | 0.23 | 0.09 | 1.25 | 99.5 |
| 2017 | 0.29 | 0.09 | 1.86 | 86.0 |
| 2018 | 0.34 | 0.11 | 1.76 | 99.3 |
| 2019 | 0.31 | 0.12 | 1.73 | 99.1 |
| 2020 | 0.22 | 0.10 | 1.41 | 95.9 |

* Low $N$ (%) is attributed to the removal of data from 22 May 2010 to 15 June 2011 due to abnormally low values, likely caused by overly strict data processing.

**Table A3.** Annual average and median concentrations of $NO_x$ on Utö. Values for the period 2003–2005 are based on hourly averages, while, for the period 2006–2020, 1 min time-resolution data are used. STD is the standard deviation. $N$ (%) represents the fraction of the year for which high-quality data are available, expressed as a percentage. High-quality data, used in our analysis, are defined as valid data recorded, excluding those compromised by factors such as instrument malfunctions, environmental interference, or calibration issues.

| $NO_x$ | Mean ($\mu g\,m^{-3}$) | Median ($\mu g\,m^{-3}$) | STD ($\mu g\,m^{-3}$) | $N$ (%) |
|---|---|---|---|---|
| 2003 | – | – | – | – |
| 2004 | – | – | – | – |
| 2005 | – | – | – | – |
| 2006 | – | – | – | – |
| 2007 | – | – | – | – |
| 2008 | 2.57 | 1.51 | 4.43 | 47.1 |
| 2009 | 3.32 | 2.12 | 4.46 | 98.1 |
| 2010 | 4.20 | 2.72 | 5.66 | 35.9* |
| 2011 | 3.41 | 2.39 | 4.09 | 50.2* |
| 2012 | 3.87 | 2.59 | 4.93 | 98.1 |
| 2013 | 3.69 | 2.46 | 4.75 | 98.0 |
| 2014 | 4.32 | 2.98 | 4.97 | 99.4 |
| 2015 | 3.31 | 2.13 | 4.35 | 98.6 |
| 2016 | 3.47 | 2.25 | 4.91 | 99.5 |
| 2017 | 3.50 | 2.13 | 5.97 | 86.0 |
| 2018 | 4.49 | 3.01 | 5.92 | 99.3 |
| 2019 | 4.06 | 2.59 | 5.91 | 99.1 |
| 2020 | 3.27 | 2.31 | 4.78 | 95.9 |

* Low $N$ (%) is attributed to the removal of data from 22 May 2010 to 15 June 2011 due to abnormally low values, likely caused by overly strict data processing.

**Table A4.** Annual average and median concentrations of $O_3$ on Utö. Values for the period 2003–2005 are based on hourly averages, while, for the period 2006–2020, 1 min time-resolution data are used. STD is the standard deviation. $N$ (%) represents the fraction of the year for which high-quality data are available, expressed as a percentage. High-quality data, used in our analysis, are defined as valid data recorded, excluding those compromised by factors such as instrument malfunctions, environmental interference, or calibration issues.

| $O_3$ | Mean ($\mu g\,m^{-3}$) | Median ($\mu g\,m^{-3}$) | STD ($\mu g\,m^{-3}$) | $N$ (%) |
|---|---|---|---|---|
| 2003 | 67.04 | – | 19.97 | 92.3 |
| 2004 | 69.73 | – | 16.47 | 90.0 |
| 2005 | 68.71 | – | 18.20 | 97.1 |
| 2006 | 74.94 | 74.08 | 22.23 | 77.6 |
| 2007 | 65.66 | 65.87 | 16.46 | 99.8 |
| 2008 | 64.99 | 63.46 | 20.38 | 98.8 |
| 2009 | 60.09 | 59.91 | 16.90 | 98.3 |
| 2010 | 64.43 | 64.49 | 16.35 | 99.1 |
| 2011 | 66.17 | 66.10 | 16.42 | 95.1 |
| 2012 | 65.64 | 66.28 | 16.00 | 97.5 |
| 2013 | 70.28 | 68.93 | 18.05 | 98.4 |
| 2014 | 62.36 | 62.43 | 15.98 | 82.0 |
| 2015 | 67.58 | 68.13 | 13.65 | 99.3 |
| 2016 | 66.68 | 66.51 | 15.86 | 99.5 |
| 2017 | 65.61 | 66.83 | 14.53 | 98.7 |
| 2018 | 67.64 | 67.59 | 17.67 | 99.3 |
| 2019 | 67.25 | 67.72 | 17.92 | 99.1 |
| 2020 | 64.55 | 65.35 | 15.55 | 99.1 |

**Code availability.** Codes for the analysis of long-term pollutant concentrations are available upon request from Androniki Maragkidou (androniki.maragkidou@fmi.fi). Codes for the concentration analyses are available from Tiia Grönholm (tiia.gronholm@fmi.fi).

**Data availability.** The 1 min air quality data from 2006–2020 are available on Zenodo (https://doi.org/10.5281/zenodo.14500222; Maragkidou et al., 2024). The dataset also includes 10 min resolution meteorological data (wind speed, wind direction, air temperature, air pressure, relative humidity, and precipitation) and 1 h air quality data from 2003–2005. Meteorological data are also available from FMI Open Data (https://en.ilmatieteenlaitos.fi/open-data, FMI, 2025).

**Author contributions.** AM conducted part of the analysis, wrote the first draft of the article, and led the processing of the article to its final format. TG performed most of the data analysis and part of the writing. LR processed the AQ data and computed some of the figures. LR, LL, and JK contributed to the planning of the study and the writing of the article. TM and TA were responsible for the measurements on the island of Utö and contributed to the analysis of the measurements. J-PJ did the computations on the emissions of shipping using STEAM. JN is responsible for the computer anal-

yses underlying Fig. 7. All authors commented on the article and provided feedback.

**Competing interests.** The contact author has declared that none of the authors has any competing interests.

**Disclaimer.** Publisher's note: Copernicus Publications remains neutral with regard to jurisdictional claims made in the text, published maps, institutional affiliations, or any other geographical representation in this paper. While Copernicus Publications makes every effort to include appropriate place names, the final responsibility lies with the authors.

**Acknowledgements.** The authors would like to thank Marko Juola for supporting the measurements.

**Financial support.** This work was partly funded by the European Union's Horizon 2020 research and innovation programme: Evaluation, control and Mitigation of the EnviRonmental impacts of shippinG Emissions (EMERGE) (grant no. 874990). This work reflects only the authors' views, and CINEA is not responsible for any use of the information it contains. Part of the research has been funded through H2020 project Joint European Research Infrastructure of Coastal Observatories: Science, Service, Sustainability (JERICO-S3) (grant no. 871153). Observations carried out on Utö have utilized the Integrated Carbon Observing System (ICOS); Aerosol, Clouds and Trace Gases Research Infrastructure (ACTRIS); and Finnish Marine Research Infrastructure (FINMARI) research facilities and resources.

**Review statement.** This paper was edited by Lisa Whalley and reviewed by Mingxi Yang and one anonymous referee.

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

## Remarks from the typesetter

**TS1** **Dear Editor: the following sentence should be changed to: On a European scale, the European Environment Agency and European Maritime Safety Agency (EEA and EMSA, 2025) evaluated the maritime sector's share of EU–27 transport SOx emissions from 1990 to 2022 and indicated that the share increased steadily, peaking at 96 % in 2005, remained stable at 94%–96 % until 2019 and then declined to 88 % in 2022. The share of NOx emissions from the maritime sector in total transport emissions in the EU-27 has steadily increased over time, reaching 39 % of all transport-related $NO_x$ emissions in 2020–2022. From 2015 to 2023, $NO_x$ emissions rose by 33 % in the Atlantic and 8 % in the Mediterranean Sea, whereas they declined by 17 % in the North Sea, 7 % in the Black Sea, and 6 % in the Baltic Sea. Additionally, the maritime sector's share of total $PM_{2.5}$ emissions from transport has gradually risen over time, peaking at 43 % in 2018 and 2019 and reaching a similar level in 2022.**