# Peer review of "Measurement Report: The effects of SECA regulations on the atmospheric SO2 concentrations in the Baltic Sea, based on long-term observations at the Finnish Utö Island."

_EGUsphere, 2024_

## Author Response (AR1)

**Response to reviewers' comment on "Measurement Report: The effects of SECA regulations on the atmospheric SO2 concentrations in the Baltic Sea, based on long-term observations at the Finnish Utö Island." by Maragkidou et al., EGUSPHERE2024-1703**

**Response to Reviewer #1 comments**

*For clarity, our response to the reviewer's comment is in **purple** font.*

This paper reports on the long term (2003-2020) measurements of atmospheric trace gases and aerosols from an island in Finland, with a particular focus on the impact of shipping regulation on SO2. The authors show that the SO2 concentrations have decreased significantly, especially since 2015, while NOx, PM2.5 haven't really changed. This is clearly a very valuable timeseries dataset, thanks to its length. However at present I don't feel like I've learned much new from reading this paper. There have been earlier studies that report on the reductions in atmospheric SO2 in coastal areas following the IMO regulations, which this paper fails to acknowledge. Other questions that could be (but not currently) addressed this paper include:

- Fuel sulfur content and the rate of compliance by ships
- Does the observed trend in SO2 reflect what one expects (e.g. based on atmospheric transport modelling with STEAM emission and terrestrial S sources)?
- Atmospheric processing of trace gases

**Reply:** 1) We would like to thank the reviewer for his comment. However, as we do not have simultaneous $CO_2$ measurements next to $SO_2$ measurements, we were unfortunately not able to analyze fuel sulfur content and the rate of compliance with legislation implemented.
2) Figure 6 presents the observed $SO_2$ to $NO$ concentration ratios, demonstrating a significant reduction in $SO_2$ levels following the implementation of SECA regulations in 2015. This decreasing trend aligns with Figure 3, which shows a clear downward trajectory in $SO_x$ emissions not only from the shipping sector but also across various other sectors (e.g., public power, industry) in countries bordering the Baltic Sea, including Finland and, consequently, the Utö site. Additionally, this trend is consistent with the predicted $SO_x$ emissions from maritime traffic, as shown by the STEAM model in Figure 2.
Moreover, while employing a chemical transport modelling would be useful, it's important to note that this was not within the scope of this study. However, we understand the importance of such modeling in interpreting our findings, thus our data is available for those who wish to conduct this kind of analysis.
3) Regarding point 3, we would like to point out that the shipping lane is very close to the measurement location and the emissions reach the measurement location in under 10 minutes (according to now former Table 3). Additionally, considering that atmospheric conditions such as $O_3$ levels and solar radiation have remained relatively stable over the past two decades, we believe that atmospheric processing doesn't have a major impact on $SO_2$ concentrations, and observed changes in close-by plumes we measured.

Specific comments

Line 66. This paragraph is ok in isolation. But it's not something this work will address. It's probably better to remove or significantly shorten it.

Instead, before line 79 it would be useful to review previous work on ship SO2 emissions (including time series measurements similar to this study, e.g. doi:10.5194/acp-15-5229-2015 and http://www.atmos-chem-phys.net/16/4771/2016/). What are the knowledge gaps that this paper can fill? Is it e.g.
- impact of regulation on atmospheric pollutant level?
- scrubber vs. low sulfur fuel?
- rate of compliance from ships?
- attribution of emission to different ship types?
- atmospheric processing and transformation of trace gases
**Reply:** Thank you for the helpful feedback. Regarding the paragraph at line 66, our goal was to underscore that while the SECA regulations were primarily introduced to protect human health by reducing the sulfur content in fuel, they've also had some unintended consequences on climate and marine ecosystems. However, we have significantly shortened the paragraph and moved it to a more appropriate place within the introduction section to improve the flow and the transition from one paragraph to the next. We also added the papers the reviewer suggested (Yang et al., 2016; Beecken et al., 2015), including three more references. The introduction and the references sections were, therefore, revised.
We also feel that our study fills an important gap by providing a detailed, long-term analysis of air quality trends in the Baltic Sea, since to our knowledge, there are no other published long-term $SO_2$ datasets that have been collected in the middle of the sea for the Baltic Sea. We hope this work, together with the dataset described in this paper, and made available will contribute valuable insights into the effectiveness of SECA regulations and serve as a benchmark for local and regional scale dispersion modeling.
We have revised accordingly the Conclusions section to address the knowledge gaps that this work fills in and explain its novelty in comparison to similar studies.

4.1 there is nothing wrong with this section on its own. However I don't feel like it contributes very much to the paper at the moment. I guess the key message is that S emission from non-shipping sectors has been declining gradually, while S emission from shipping has decreased in step wise fashion following the IMO regulations (which we would've expected even without STEAM model)?
I think this section can be made more powerful if the authors implement these emissions in an atmospheric transport model, see what the predicted change in SO2 is, and then compare the model with the observations.
**Reply:** We thank the reviewer for his suggestion. However, the focus of this paper is on observed time series with some supporting data. Hence, employing an atmospheric transport model is outside of this study's scope. As the data will be available for modelers, we are really interested in seeing in the future outcomes of the modeling studies utilizing our observations

Line 158. SO2 lifetime was estimated to be only 0.5 day to the west of the UK. http://www.atmos-chem-phys.net/16/4771/2016/ in cloud oxidation is probably the largest sink.
**Reply:** Lee et al. (2011) and Beirle et al. (2014) evidenced that the average lifetime of SO2 in the lower atmosphere is ~1–3 days. However, this can vary according to atmospheric conditions, temperature and humidity. The duration of $SO_2$ lifetime doesn't have an impact on the SECA regulations. The above-mentioned references (i.e. Lee et al. (2011) and Beirle et al. (2014)) were added to this line to support the statement as well as to the references section.

Fig.2 how come the SOx emission from shipping sector hasn't decreased in step-wise fashion, corresponding to the regulations? Is it because the emission also includes outside of SECA?
**Reply:** We would like to point out to the reviewer that the y-axis in Figure 2 is on a logarithmic scale, thus making the gradual reductions in $SO_x$ emissions probably less visually apparent. Moreover, as it is stated in the figure caption, Figure 2 demonstrates the combined annual $SO_x$ emissions from Estonia, Finland, Lithuania, Latvia, Sweden, Russia, and Poland, all of which are within the SECA. Therefore, it does not include emissions from outside the SECA, ensuring that the data reflects the impact of SECA regulations alone. However, Figure 2 was revised to enhance its resolution and the following statement "*Note that the figure is presented on a*

*logarithmic scale (y-axis). The source of SOx emissions data for the different sectors is EMEP (European Monitoring and Evaluation Programme).*" was added to the figure caption.

In addition, Figure 3 shows clearly the step-wise reductions of $SO_2$ following the introduction of each SECA regulation.

Table 1. a bit more detail on the SO2 measurements would be useful (even if this info had been reported previously elsewhere). E.g. how was it blanked and calibrated?

**Reply:** Per reviewer's request, we added information on how instruments were blanked and calibrated to subsection 3.1, below Table 1.

Line 169-170 this is a repeat

**Reply:** We would like to thank the reviewer for his observation. We have deleted this sentence as it was repeated earlier.

Line 176. 'until'

**Reply:** We would like to inform the reviewer that the entire text below Figure 2 has been revised, including the replacement of 'till' with 'until' in the manuscript.

Table 2. what's N(%)?

**Reply:** N (%) represents the fraction of the year for which high-quality data is available, expressed as a percentage. High-quality data, used in our analysis, are defined as valid data recorded, excluding those compromised by factors such as instrument malfunctions, environmental interference, or calibration issues. We have added the definition of N (%) to the caption of Table 2 and Tables A1-A4.

Figure 5. how come percentiles and median are not shown for data over the first few years? Is it because the data were hourly, not minutely? In general, I don't really see the value of presenting/analyzing minutely data for this section. Hourly data would be perfectly fine for looking at long term trends. Minutely data contain much more measurement noise, especially for SO2.

**Reply:** We would like to clarify that for the first period the data is only hourly and, thus, 1st and 99th percentiles for this period are not comparable with the second period with 1-minute resolution data. Analysis of nearby plumes requires higher time resolution data since typically the plumes from the ships are few minutes in duration; therefore, we have employed 1 minute data. In addition, 1-minute data is also better suited for possible later local and regional transport modelling studies.

Line 238. How were 'peaks' identified/defined? What's the minimum NO concentration in this calculation? Have you accounted for any possible lag between the SO2 and NO data due to imperfect time synchronization or different instrument response times?

**Reply:** Peaks were identified as a sudden and simultaneous increase of NO and $SO_2$ concentrations. The minimum peak prominence was set to be 2 µg m$^{-3}$. The different instrument response times have not been taken into account since the difference is at maximum few seconds and the data is presented in time resolution of 1 minute. The peaks in the data caused by a ship passing by Utö during wind blowing from the shipping lane towards the measurement station are typically very clear and well synchronized (see the figure below presenting a plume by a ship, not included in the manuscript). The analysis shows qualitatively that the $SO_2$ peaks caused

by near-by-sources (ships) are significantly lower after SECA2015 compared to earlier years.

[Figure]

Figure 6. NO reacts rapidly with O3 and has a strong diurnal cycle and SO2:NO will too. Was CO2 not measured at the site, which would've enabled the estimation of the fuel sulfur content? If not, SO2:NOx still seems better than SO2:NO, though NOx emissions can vary significantly depending on the ship/weather conditions.

**Reply:** As we mentioned earlier, $CO_2$ measurements were not available for the whole studied period at the same location. $CO_2$ measurements have been available from the marine site since March 2014. Given this, we focused on analyzing the relationship between NO and $SO_2$. If both NO and $SO_2$ show concurrent peaks, it suggests a nearby emission source rather than long-distance transport, as NO would be rapidly oxidized to $NO_2$ over time. Therefore, we examined the $SO_2$:NO ratio for specific ships that operated throughout the study period from 2006 to 2020. Any significant changes in this ratio would imply shifts in fuel sulfur content, likely due to regulatory changes over the years.

Section 4.1 it's a bit odd to have this section here, when you just chosen the western sector (180-360) for the SO2:NO analysis above. Wouldn't be better to do the wind sector analysis first, and then apply the according wind sector to SO2:NO?

**Reply:** We thank the reviewer for his feedback. However, we respectfully disagree with this suggestion, as we chose the current structure in order to establish a logical flow and clarity of our results and discussion. Therefore, we decided to keep the structure as it is.

Figure 7. similar to figure 5, the SO2 axis is cut off at zero. Are you discarding all negative SO2 data? I don't think that is the best approach. The negative numbers (due to measurement noise) need to be kept in in order for the stats to be representative.

**Reply:** We would like to inform the reviewer that Figure 7 was revised to include some statistical information, as well. Moreover, we would like to clarify that none of the measured SO₂ concentrations, including any negative values due to measurement noise, were discarded during the analysis. All measurement data were applied in the creation of the figure.

However, we chose to restrict the vertical axis of Figure 7 to a range of 0-80 µg m⁻³ for better visual clarity (the range of the measurement data was from -3.5 to 178 µg m⁻³). As a result, a negligible portion of the 1-minute measurement data, including negative values, was not shown on the figure. This decision was made for visualization purposes only and does not affect the statistical representation of the data, as the mean and median values, newly introduced in the revised figure, were calculated using the complete dataset, including negative values.

Figure 9. I don't doubt that the SECA regulation has been effective. However here the SO2 and NOx concentrations were not evaluated with consideration of plume dilution. Would've been best to normalize both gases to CO2 plume. If that's not possible, at least look at SO2:NOx ratio (which does seem to be lower after 2015).

**Reply**: We thank the reviewer for his suggestion, but after careful consideration and per his request, we decided to remove subsection 4.4 and thus Figure 9.

In general, I don't find that the case study has added much to the paper. Are there more information that can be teased out? E.g. fuel sulfur content before vs after 2015? Did the ship install a scrubber?

**Reply**: We thank the reviewer for his feedback. Therefore, after careful consideration and per his request, we decided to remove the case study part (i.e. subsection 4.4).

**Response to Reviewer #2 comments**

*For clarity, our response to the reviewer's comment is in* ***purple*** *font*

Review of the preprint "Measurement Report: The effects of SECA regulations on the atmospheric SO2 concentrations in the Baltic Sea, based on long-term observations at the Finnish Utö Island" by A. Maragkidou et al., submitted to Atmospheric Chemistry and Physics

The manuscript describes a long term observation of air pollutants on the Finnish Utö island. The analysis is focused on SO2 concentrations in order to demonstrate the effects of SECA regulations to reduce sulphur emissions from shipping. It is an interesting paper that deserves publication, but improvements and clarifications are needed.

Major general comments:

It isn't exactly clear how you use the STEAM data in your analysis. Please include it in the data interpretation or leave it out.
**Reply:** As requested, we have added a couple of lines to subsection 3.3, in which we describe how STEAM data was used in the analysis.

I am not convinced that we can learn something from the RoRo ship case study (section 4.5). Please explain this better in the paper or remove this part.
**Reply:** We thank the reviewer for his feedback. Therefore, after careful consideration and per his request, we decided to remove subsection 4.5.

Please include a paragraph/section about the limitations of your study and about the uncertainties. This should include a discussion about the representativity of the observations for a larger region. Can we really say something about the compliance to the SECA rules in the Baltic Sea when we have approx. 8 ships passing by per day?
**Reply:** Firstly, we would like to inform the reviewer that Figure 4 was revised to include more information on the number of ships that pass Utö on a daily and yearly basis, with many of these vessels passing multiple times per day. Moreover, the purpose of this study was not to assess the compliance of ships with SECA regulations, but to assess the effectiveness and effects of the impact of SECA regulations on $SO_2$ concentrations in the Baltic Sea. We recognize that this part wasn't clear in the introduction, therefore we revised the introduction to reflect that point.
Regarding uncertainties, given the nearly two-decade observation period, several factors could introduce uncertainties in the results. Instrumentation for all variables changed over time, and quality assurance methods improved. Additionally, there were changes in the personnel responsible for maintaining the instruments and ensuring data quality. Although standard protocols and measurement diaries were followed, these transitions may have affected the results.
While our observations are important for long–term, high time resolution air quality observations at remote marine research stations, in the vicinity of a heavily trafficked ship lane, as already mentioned in the manuscript, they do not directly represent the broader Baltic Sea region.
Per the reviewer's request, we have addressed the uncertainties of this study in a new subsection "4.4 Uncertainties" of the manuscript.
In terms of limitations, our dataset offers unique fine temporal resolution data of air quality observations at a remote marine station near a heavily trafficked shipping lane. However, the data collected from 2003-2005 were recorded at hourly intervals, whereas the data from 2006-2020 were gathered at 1-minute intervals.

Additionally, $CO_2$ data was not available for the whole period for the same location and therefore, we were unable to estimate fuel sulfur content. Moreover, no local scale dispersion modelling was employed as it was outside of this study scope, However, the dataset of this work could serve as a benchmark for local and regional scale dispersion modelling. Utö itself is a sparsely populated island with minimal land traffic and no significant local sources of NOx and SO2.

Major specific comments:

Line 135-137: Which data was used for the STEAM model (2006 – 2020)? Why was there a change in AIS data source for the RoRo ship in 2016?
**Reply:** For the case study, we used AIS data from the AIS receiver installed in Utö in 2015. Therefore, for the period 2016-2019 AIS data from AIS receiver at the Utö marine station was used. For earlier years (2013-2015), HELCOM data set was used. Furthermore, AIS data prior to 2013 was not accessible to us. However, per reviewer's request, subsection 4.5 and hence these lines were removed. For the STEAM model, the AIS data used was from HELCOM AIS for the Baltic Sea since 2006.

Line 151/152 and section 4.5: I do not see the purpose of the case study looking at one RoRo ship. What do you want to demonstrate? What can we learn from this?
**Reply:** We thank the reviewer for his feedback. Therefore, after careful consideration and per his request, we decided to remove subsection 4.5.

Line 164/165: What is the data source for this graph?
**Reply:** The data source for Figure 2 is EMEP (European Monitoring and Evaluation Programme). The following clarification was also added to the Figure caption "*The source of SOx emissions data for the different sectors is EMEP (European Monitoring and Evaluation Programme)*".

Line 179/180 and Fig. 3: Why are PM2.5 emissions only available with one decimal place?
**Reply:** We did not provide any numerical values for $PM_{2.5}$ emissions in lines 179-180. The $PM_{2.5}$ emissions shown on the right y-axis of Figure 3 are displayed with one decimal place as the default setting.

Line 180 and Fig. 3: What happens with the NOx emissions in 2017? After a steady decrease until end of 2017 something looks different in the data with an increase in 2018. It think you cannot say that the emissions remained stable. It would also be interesting to see the CO2 emissions from STEAM in order to see the effect of increased ship traffic vs. more efficient fuel use. Besides the sulphur content in ship fuels, the total amount of fuel burned has an influence on SO2 concentrations. This is not much discussed in the paper.
**Reply:** Thank you for your comment. Regarding the $NO_x$ emissions in 2017, we observed a significant reduction—approximately 95%—in the amount of AIS data received starting from 6 November 6 2017 until 31 December 31 2017. This reduction appears to be an issue with AIS data reception rather than with data storage, as the decline is evident in both the HELCOM AIS and global AIS datasets. Therefore, it is unlikely that this discrepancy is related to HELCOM's data storage system. The following statement "*except for between 6 November 2017 and 31 December 2017, when a 95% decrease in AIS data was observed due to an issue with AIS data reception*" was added to the end of this sentence in subsection 4.1: "*The predicted annual NOx emissions during the period between 2006 and 2020 have remained relatively stable*" to explain why the significant reduction that was observed during that period.

Fig. 5, b): Why does the median for PM2.5 reach 0 between 2016 and 2018? It looks like the median and the mean are lower in 2015-2017 and then go up again in 2018. Is this connected to a change in instruments?

**Reply:** Figures 5a-d were revised by filtering out extreme peaks and negative values identified as invalid data, likely caused by changes in instrumentation and improvements in quality assurance methods over time. $PM_{2.5}$ specifically had some negative values that were removed in the revised version of Fig. 5b, resulting in gaps in the time series. While all changes followed standard protocols and were documented in measurement diaries, shifts in equipment and personnel may have introduced some variability. We have carefully accounted for these factors and performed additional quality checks to ensure the integrity of the data presented. Moreover, the following sentence was added to subsection 4.2, below Table 2: "*$PM_{2.5}$ concentrations also had some negative values, which were removed, leading to gaps in the time series (Fig. 5b)*." to justify the gaps observed during some periods.

Fig. 5, c): It looks like NO is lower since 2014, this holds in particular for the median. This is also visible in the data in the appendix and it is in contrast to what you say in the text. Can you comment on it? Is it because of the change in instruments? Could you please elaborate on the effects of instrument changes in general?
**Reply:** As previously mentioned, Figures 5a-d were revised by filtering out extreme peaks and negative values, which were identified as invalid data, likely due to changes in instrumentation and the changes in quality assurance protocols over time.
Specifically for NO (and $NO_x$), data from the period 22.5.2010 to 15.6.2011 were excluded in the revised version of Fig. 5c (as well as of Fig. 5d) due to abnormally low values, potentially caused by overly strict data processing during that time. Additionally, since the medians for hourly data (2003-2005) are not directly comparable to the medians for 1-minute data (2006-2020), we have removed the medians for the 2003-2005 period from Tables 2 and Tables A1-A2 and A4 to ensure consistency and accuracy in the data presented.
The following sentence: "*However, for NO and $NO_x$, a period of data from 22 May 2010 to 15 June 2011 was removed (Figs. 5c and 5d) due to abnormally low values, likely caused by overly strict data processing*." was added to subsection 4.2, below Table 2, to justify the gaps observed during this period.

Fig. 6: Why is there no data in 2011? And could you give some statistical information? How large is the reduction in 2015-2020 compared to the earlier years? And how does this compare to the expected reduction because of more stringent SECA rules.
**Reply:** As mentioned earlier, there was no NO data during the period 22.5.2010 to 15.6.2011 due to abnormally low values, potentially caused by overly strict data processing during that time. In addition, we didn't employ a chemical dispersion model in our study; therefore, we couldn't calculate and compare the expected reduction resulting from the implementation of stricter SECA regulations.

Line 155 and Fig. 7: Please enhance the figures with some statistical information or use a separate table for this. How does the data compare to that from all wind sectors?
**Reply:** As requested, we have revised Figure 7 by adding the time series of moving mean and moving median. The figure's caption was accordingly revised, as well.

Line 267: It is not clear why you select 2019 as the year with lower sulphur emissions from shipping. Why not 2016/17/18? Having possible changes in emissions from other sources in mind, 2019 seems to be not the preferred choice.
**Reply:** In terms of $SO_x$ emissions from shipping, the years 2016, 2017, 2018 and 2019 are not substantially different, as can be seen based on Fig. 2. The reviewer probably refers here to the impacts of COVID-19, which is of course a relevant point. However, many studies have shown that the impacts of COVID-19 on emissions in Europe were not felt yet in 2019; these only affected the subsequent years, since 2020. For instance, the first case of COVID-19 documented in the UK was on 31 January 2020.
We have revised the manuscript to clarify why we selected the year 2019. We have added the following rationale to subsection 4.3, below Figure 7: "*The years 2016, 2017, 2018 and 2019 were not substantially different in terms of $SO_x$ emissions from shipping (cf. Fig. 2) or the number of ships (Fig 4). In this regard,*

*any of these years could have been selected as an example year for the post-SECA 2015 analysis. The selected year 2019 was prior to the COVID-19 pandemic and selected for comparison. The pandemic did not affect the emissions in Europe in 2019; these effects were felt only during the subsequent years.*"

Line 273/274 and Fig. 8 a) and b): It seems that SO2 mean and/or median does not depend very much on wind direction. Can you comment on this?
**Reply:** Thank you for your observation. However, we would like to point out that Figure 8 depicts histograms of $SO_2$ concentrations, not mean or median values. The radial axis in Fig. 8 represents the number of measured cases for each wind direction sector.
These histograms represent the measured concentrations of $SO_2$ at the island of Utö, with a time resolution of 1–minute, plotted as a function of wind direction. From Figure 8, it is clear that the dominant wind directions are southwesterly, with relatively high contributions also from northwesterly directions (Figure 8a, Figure 8b and Figure 8c).
Figures 8c-d present only the occurrence of the highest concentrations. It is very clear based on panel c that a vast majority of the highest measured concentrations in 2014 originated from the direction of the shipping lane (which was west of the island).
Moreover, a significant difference is observed between the years 2014 and 2019, suggesting that the implementation of SECA regulations has effectively reduced the number of high concentration peaks (Figure 8c and Figure 8d).

Fig.8 a) and b): The reader has the impression that violet values are $0 – 0.5$ µg/m³, yellow values are $0.5 – 1.5$ µg/m³, and red values are $>1.5$ µg/m³. This is in contrast to the legend and the caption.
**Reply:** The figure caption was slightly revised to clarify what each color represents and how it was plotted. It now reads as follows: *"Figure 8: The measured concentrations of $SO_2$ at the island of Utö as a function of the wind direction, as polar histograms with 1-minute resolution. The panels a) and c) correspond to the data in 2014 and the panels b) and d) to those in in 2019. These two years were selected to represent the situation before and after the SECA regulation in 2015. The upper panels (a and b) include all datapoints, whereas the lower panels (c and d) include only the data, for which the concentrations were higher than a selected threshold value (5 µg m⁻³). In panels a) and b), the red colour in the legend represents all data points with values larger than 0 µg m⁻³. Overlaid on the red, the yellow colour shows data points with values below 1.5 µg m⁻³, and the violet colour, which overlays the yellow, indicates data points with values below 0.5 µg m⁻³. The radial axis represents the number of measured cases for each wind direction sector.*"

Lines 286-329, Section 4.5: As said before, this section does not provide new insights. You may completely skip it unless you describe better, what is new and what can be learned from it.
**Reply:** We thank the reviewer for his feedback. Therefore, after careful consideration and per his request, we decided to remove subsection 4.5.

Line 335: The STEAM results are not used very much. You should improve this.
**Reply:** The purpose of this paper was to present the long-term time series of $SO_x$ measurements collected at a marine monitoring station located near a major shipping lane. While we acknowledge the importance of modeling, our focus was on measurements, with modeling serving to complement and enhance the interpretation of the time series. Specifically, the STEAM data was employed to support the analysis by providing information on ship traffic, such as the number of vessels, and to examine the temporal trends in the measured $SO_x$ concentrations.

Line 339/340: You would underpin this statement with a trend analysis including statistical significance of the trend. It is also in contrast to steady emission reductions in many emission sectors in Europe (Fig. 2). Therefore, you may a few words on this.

**Reply:** Regarding the concentrations of $SO_2$, if one would make a statistical trend analysis, it should consider separately the 3 periods, separated by the SECA changes. One should address annual average concentrations, due to the changing meteorological conditions for each year. However, there are far too few annual data points in each of the 3 periods for conducting such a statistical analysis.

This study focused on $SO_2$ concentrations. A detailed statistical study of the other considered pollutants is outside the scope of this manuscript. An examination of the data shows that there were no substantial trends for the other pollutants. Please note also that Fig. 2 in the manuscript presents only the emissions of $SO_x$ and Fig.3 presents emissions of $SO_x$, $PM_{2.5}$ and $NO_x$, but only those from shipping.

Line 340/341: This could also be caused by changes or variations in emissions

**Reply:** We agree with this point and have improved the text in the Conclusions section to read: *"The year–to–year variations of the concentrations were substantial for all pollutants; these were attributed partly to the variations in regional meteorology, partly to the variations of emissions"*.

Line 350/351: I would like to read some words about the possibilities to check compliance to the NECA since 2021. What are the prospects for the future for these observations related to air pollution from shipping?

**Reply:** We thank the reviewer for the suggestion. However, checking compliance with the NECA regulations since 2021 could be the focus of a separate, new study. As highlighted in our paper, the primary objective of this work was to assess the effects and effectiveness of SECA regulations on atmospheric $SO_2$ concentrations in the Baltic Sea, as well as on $PM_{2.5}$, $NO_x$, NO, and $O_3$ concentrations, using long-term air quality data measured at the Utö Island, and not assessing the rate of compliance of ships with the SECAs regulations implemented. In addition, monitoring $NO_x$ compliance is more challenging due to the regulations being based on engine power output, which set emission limits based on engine performance.

The dataset of this work is particularly unique because there are no other similar long-term air quality datasets from remote locations in the Baltic Sea, and it has not been previously published or analyzed.

Moreover, Utö Island's location in the middle of the Baltic Sea makes it an ideal site for studying long-term pollution trends from shipping, with minimal influence from other pollution sources. Our work underscores the significance of long-term, high-resolution air quality monitoring at remote marine research stations, especially those near heavily trafficked shipping lanes. Such observations are crucial for both quantitative and qualitative analyses of the impacts of regulatory environmental regulations.

Minor comments:

Line 98: Is there really no local wood burning that may have an effect on SO2 and PM2.5 concentrations? Or were these events removed from the data?

Reply: As we mentioned in the manuscript, Utö is a small island in the Baltic Sea, with less than 1 km². It is located 70 km off the coast of mainland Finland, and it is surrounded by open sea and a few smaller islands. During the winter, less than 40 inhabitants live in Utö. Because Utö is so sparsely populated and has very little land traffic, there aren't any significant local sources of $NO_x$ or $SO_2$. Although some firewood is used for heating, it has to be imported, and its contribution to $SO_2$ emissions is minimal. While local wood burning might have a slight effect on $PM_{2.5}$, it's unlikely to have a major impact on the long-term trends we've observed in our study.

Line 131: These vessel categories are not well defined. What is "small" and what is "large". Please give more details. And why is there no further distinction of cargo ships into e.g. container ships, tankers, bulk cargo, …? I would assume that STEAM considers more categories than six.

**Reply:** Vessels with an IMO registry number are classified as "large," while smaller vessels transmit only an MMSI code and no IMO number. The primary aim of this paper is to demonstrate the reduction in $SO_x$ concentration trends, regardless of vessel type.

Line 158: Correct: "is therefore originating"
**Reply:** The sentence was corrected to "*Part of the $SO_2$ observed at Utö therefore originates from long-range transport of regional background pollution, while another portion is attributed to shipping traffic in the vicinity of Utö.*", since grammatically is more correct.

Line 173: Effects of the pandemic on emissions were not visible before 2020.
**Reply:** According to HELCOM (2021), air pollutant emissions from the Baltic Sea ship fleet have declined since 2019, largely due to the COVID-19 pandemic, which significantly decreased vessel activity in the region.

Line 190/191: "The amount of ship traffic has been fairly constant during this period." repeats what was said before.
**Reply:** After careful consideration we decided to delete this sentence, as it repeats what was said before.

Line 205, Table 2: Please explain STD, N, TBA. Are there no units for STD?
**Reply:** STD= Standard deviation in [$\mu$g m$^{-3}$].
N (%) represents the fraction of the year for which high-quality data is available, expressed as a percentage. High-quality data, used in our analysis, are defined as valid data recorded, excluding those compromised by factors such as instrument malfunctions, environmental interference, or calibration issues.
We have added all the necessary clarifications to the respective Tables and revised them.

Line 232/233: please make clear that these changes always refer to the 2003-2005 values (and not to the previous period).
**Reply**: This sentence was changed to "*Our findings revealed that three-year average $SO_2$ concentrations from the pre-SECA period (2003–2005) decreased by 38 %, 39 %, and 67 % in comparison to the post-SECA periods (2007–2009, 2011–2013, and 2016–2018), respectively*".

Line 240/241: Perhaps you want to introduce abbreviations like SWECA2006, SECA2011, SECA2015 to make clear which phase of the regulation you talk about. This might also help at other places (e.g. line 256, but there may be more).
**Reply**: We would like to thank the reviewer for this suggestion. We have incorporated the suggested abbreviations (i.e., SECA2006, SECA2010, SECA2015) throughout the text where applicable.

Line 242: replace "during" with "within"
**Reply**: We thank the reviewer for this correction. We will replace "during" with "within" per the reviewer's recommendation.

Line 250-258: These paragraphs need improvements of the English language (e.g. articles).
**Reply:** We would like to thank the reviewer for his suggestion. We have revised the text to improve clarity and correctness, especially in terms the English language. The updated paragraphs now read: "*To study the impact of ships passing by Utö, we selected $SO_2$ concentration data based on wind direction. First, we separated the data points measured when the wind was blowing from the shipping lane (covering wind directions from 185° to 315°) towards the measurement site from the rest of the data.*

*All data, along with data from when the wind was blowing from the direction of the shipping lane and data from the background sector (wind directions excluding the shipping lane sector), are shown in Figs. 7a–c. Similar to the previously presented results, there is an evident decrease in SO$_2$ concentrations after the SECA2015, and a slight decrease after the SECA2010. This decreasing trend is visible in all three plots, but the most pronounced decrease occurs after 2015 when only wind directions from the shipping lane were considered.*"

**List of all relevant changes made in the manuscript:**

| | Changes | Date |
|---|---|---|
| | Deleted the case study part (i.e. **subsection 4.5)** and any reference to it throughout the manuscript, as suggested by the two reviewers | 15, 19 August 2024, 30, 31 October 2024 |
| | Replaced "sulphur" with "sulfur" throughout the text. | 23 October 2024 |
| | Replaced hyphen (-) with en dash (–) where appropriate according to journal's instructions | 31 October 2024, 1, 4 November 2024 |
| | **Abstract** was revised | 19 August 2024, 9 October 2024, 29 October 2024 |
| | The following abbreviations SECA2006, SECA2011, SECA2015 have been added throughout the manuscript where appropriate, as requested by Reviewer #2 | 19 August 2024 |
| | **Introduction** was revised:
i) One paragraph discussing compliance studies with SECA regulations has been added to the introduction, including two references suggested by Reviewer #1 and three additional references. | 15 October 2024 |
| | ii) The paragraph that started in old line 66 was shortened as suggested by Reviewer #1 and moved to another more appropriate place within the introduction for better flow and coherence | 23, 29, October 2024 |
| | iii) The last paragraph of the introduction has been revised and split into three smaller paragraphs to clarify the study's focus and outline the analysis conducted. | 9, 15, 22, 31 October 2024 |
| | Revisions were made to subsection "**2 Measurement location and site characteristics**":
i) Minor revisions were made to comply with journal's instructions on citations | 31 October 2024 |

| | | |
|---|---|---|
| | ii) three minor revisions were made to improve the English language and grammar | 9, 31 October 2024 |
| | Deleted a space before the last sentence of the caption of Figure 1 | 31 October 2024 |
| | Revisions wer made in "**3.1 Air quality and wind observations**": | |
| | i) Revised the first two lines of the first paragraph | 27 September 2024 |
| | ii) Added missing information for one device in Table 1 | 20 September 2024 |
| | iii) Added details on instrument blanking and calibration to subsection 3.1, below Table 1, as requested by Reviewer #1. | 19 August 2024 |
| | Revised the last paragraph of subsection "**3.2 Automatic Identification System (AIS) data**" | 14, 30 October 2024 |
| | Added a small description on how STEAM data was used in the analysis to subsection "**3.3. The STEAM model**" per Reviewer #2 request | 10, 12 September 2024 |
| | The first paragraph of Section "**4. Results and Discussion**" was revised to introduce the abbreviations SECA2006, SECA2011, and SECA2015, as requested by Reviewer #2. These abbreviations have also been incorporated throughout the manuscript. Additionally, the sentence, "*and (iii) a qualitative case study based on one regular Ro-Ro ship that has passed by the Utö Island*" was removed. | 19 August 2024 |
| | Subsection **"4.1 Changes of long-range transport and shipping emissions"** was revised**:** | |
| | i) The first paragraph of the subsection was revised to improve the English language | 9, 31 October 2024 |
| | ii) The citations Lee et al. (2011) and Beirle et al. (2014) were added to the first paragraph to support the statement on $SO_2$ lifetime, in response to Reviewer #1. | 16 August 2024 |
| | iii) The sentence (old line 158): "*Part of the $SO_2$ observed at Utö is therefore originated from long–range transport of regional background pollution,…*" was corrected to "*Part of the $SO_2$ observed at Utö therefore originates from long-range transport of regional background pollution,..*" in response to Reviewer #2' comment. | 16 August 2024, 31 October 2024 |
| | iv) Figure 2 was revised to improve its resolution in response to a comment from Reviewer #1. | 12 September 2024, 25 October 2024 |
| | v) Caption of Figure 2 was revised, and the following sentence was also added "*Note that the figure is presented on a logarithmic scale (y–axis). The source of $SO_x$ emissions data for*" | 16 October 2024, 31 October 2024, 1 November 2024 |

| | | |
|---|---|---|
| | *the different sectors is EMEP (European Monitoring and Evaluation Programme).*" as requested by Reviewer #2 | |
| | vi) The text below Figure 2 has been revised to improve coherence and better describe the results presented in the figure, including the replacement of 'till' with 'until' in the manuscript and in the old line 176, as suggested by Reviewer #1. In addition, old line 169-170 was removed as requested by Reviewer #1. Two additional citations (HELCOM, 2014; Bank of Finland, 2018) were added to the text and the following statement "*except for between 6 November 2017 and 31 December 2017, when a 95% decrease in AIS data was observed due to an issue with AIS data reception*" was added to the end of this sentence: "*The predicted annual NOx emissions during the period between 2006 and 2020 have remained relatively stable*" to explain the significant reduction observed during that period, in response to Reviewer #2. | 15, 30, 31 October 2024 |
| | vii) In response to Reviewer #2's comment, Figure 3 was revised to reflect a significant reduction—approximately 95%—in the AIS data received from November 6, 2017, to December 31, 2017. Upon reviewing our data, we found that this reduction appears to be related to an AIS data reception issue rather than a problem with data storage, as the decline is evident in both the HELCOM AIS and global AIS datasets. | 20 September 2024, 31 October 2024 |
| | viii) Caption of Figure 3 was revised to improve coherence | 24, 27 September 2024, 31 October 2024 |
| | Subsection "**4.2 Observed changes in ship traffic and concentrations of SO$_2$ and other air quality parameters at Utö**" was revised**:** | |
| | i) Figure 4 was revised to include more information on the number of ships that pass Utö on a daily and yearly basis in response to Reviewer#2 | 16, 29 October 2024 |
| | ii) The text above Figure 4 was revised to describe the results shown in Figure 4, including the removal of the sentence from the old lines 190–191, as requested by Reviewer #2. The old text was removed. | 11, 16, 23 October 2024 |
| | iii) The caption of Figure 4 has been revised to align with its description and enhance coherence. | 17, 31 October 2024 |

| | | |
|---|---|---|
| | iv) Minor revisions were made in the text below Figure 4 | 19 August 2024, 14 October 2024 |
| | v) The following sentence was added to the caption of Table 2 as requested by the reviewers: "*STD is the standard deviation. N (%) represents the fraction of the year for which high–quality data is available, expressed as a percentage. High–quality data, used in our analysis, are defined as valid data recorded, excluding those compromised by factors such as instrument malfunctions, environmental interference, or calibration issues.*" | 15 August 2024 |
| | vi) Table 2 was slightly revised due to an error in the dataset. The units of STD were also added to the table as requested by Reviewer#2. | 23, 31 October 2024 |
| | vii) The text below Table 2 was slightly revived, and we added the sentences: "*$PM_{2.5}$ concentrations also had some negative values, which were removed, leading to gaps in the time series (Fig. 5b).*"…. "*However, for NO and $NO_x$, a period of data from 22 May 2010 to 15 June 2011 was removed (Figs. 5c and 5d) due to abnormally low values, likely caused by overly strict data processing.*" as a reply to Reviewer #2's comments. | 15, 31 October 2024 |
| | viii) In response to Reviewer #2' comment, Figure 5 was revised as there was an error in the data and we had to analyze it again | 15, 31 October 2024 |
| | ix) The caption of Figure 5 has been revised to align with its description and enhance coherence | 15, 29 October 2024 |
| | x) The sentence in old lines 232-233: "*Our findings revealed notable reductions in three-year average SO2 concentrations during these periods: 38% (after SECA 2006), 39% (after SECA 2010), and 67% (after SECA 2015), respectively.*" was changed to "*Our findings revealed that three-year average $SO_2$ concentrations from the pre-SECA period (2003–2005) decreased by 38 %, 39 %, and 67 % in comparison to the post-SECA periods (2007–2009, 2011–2013, and 2016–2018), respectively.*" per Reviewer #2 request. | 31 October 2024 |
| | xi) The following sentence was added to the text above Figure 6: "*Normalization using $CO_2$ concentrations would have allowed further analysis of fuel sulfur content, unfortunately such data has not been measured at Utö in a location suitable for ship $SO_2$ plume research prior to implementation of SECA in 2015.*" | 31 October 2024 |

| | | |
|---|---|---|
| | xii) Replaced "during" with "within" (old line 242) as requested by Reviewer #2. | 16 August 2024 |
| | xiii) In response to Reviewer #2's comment, Figure 6 was revised as there was an error in the data and to improve its resolution (there was no NO data during the period 22.5.2010 to 15.6.2011 due to abnormally low values, potentially caused by overly strict data processing during that time) | 24 October 2024 |
| | Subsection "**4.3. Dependence of concentrations on local wind direction**" was revised: | |
| | i) Improvement of the English language was made in the old lines 250-258 as requested by Reviewer#2 | 19 August 2024, 14, 31 October 2024 |
| | ii) Figure 7 has been revised to enhance its resolution and to include the time series of the moving mean and moving median, as requested by Reviewer #2 | 18 August 2024, 29 October 2024 |
| | iii) The caption of Figure 7 has been revised to align with its description and enhance coherence | 18 August 2024, 31 October 2024 |
| | iv) We added the following rationale to old line 267: "*The years 2016, 2017, 2018 and 2019 were not substantially different in terms of $SO_x$ emissions from shipping (cf. Fig. 2) or the number of ships (Fig 4). In this regard, any of these years could have been selected as an example year for the post-SECA 2015 analysis. The selected year 2019 was prior to the COVID-19 pandemic and selected for comparison. The pandemic did not affect the emissions in Europe in 2019; these effects were felt only during the subsequent years.*" to clarify why we selected the year 2019, as requested by Reviewer #2 | 13 September 2024, 9, 31 October 2024 |
| | v) Figure 8 was revised to improve its resolution | 29, 31 October 2024 |
| | vi) The caption of Figure 8 was slightly revised to clarify what each color represents and how it was plotted, as requested by Reviewer #2 | 24 September 2024 |
| | A new subsection "**4.4. Uncertainties**" was added as requested by Reviewer #2 | 23, 29 October 2024 |
| | "**Conclusions**" section was revised: | |
| | i) The first paragraph was revised to improve coherence and clarity | 19 August 2024, 3,9 October 2024 |
| | ii) Old lines 340-341 were revised to: "*The year–to–year variations of the concentrations were substantial for all pollutants; these were attributed partly to the variations in regional meteorology, partly to the variations of emissions.*", as requested by Reviewer #2. | 13 September 2024, 31 October 2024 |

| | | |
|---|---|---|
| | iii) The last paragraph was revised to address the knowledge gaps that this work fills in and explain its novelty in comparison to similar studies, as requested by Reviewer #1. | 19 August 2024, 9 October 2024 |
| | Revisions were made in the "**Appendix A"** section**:**
i) Tables A1–A4 were revised as there was an error in the data
ii) Captions of Figures A1–A4 were revised to include information of STD and N (%) as requested by the reviewers
iii) Footnotes were added below Tables A2 and A3 for clarification |

23 October 2024

15 August 2024, 31 October 2024, 4 November 2024

23 October 2024 |
| | The following sentence "*and of individual ship plumes in case of one selected ship prior and post SECA*" was deleted from the "**Code availability**" section | 23 October 2024 |
| | The old text below "**Data availability**" section was replaced with: "*The 1–minute air quality data from 2006-2020 is available in Zenodo [DOI to be added]. The data set also includes 10–minute resolution meteorological data (wind speed, wind direction, air temperature, air pressure, relative humidity and precipitation) and 1-hour air quality data from 2003-2005. Meteorological data is also available from FMI Open Data (https://en.ilmatieteenlaitos.fi/open-data).* " | 29 October 2024 |
| | The following sentence was added within the text below "**Author contributions**" section: "*LR processed the AQ data and computed some of the figures.*" | 23 October 2024 |
| | Revisions were made in the "**References**" section:
i) 8 new references were added
ii) Minor revisions were made to align with the journal's style and sorting guidelines. |

16 August 2024, 24, 30 October 2024
31 October 2024 |

---

## Author Response (AR2)

**Response to Editor's comment on "Measurement Report: The effects of SECA regulations on the atmospheric SO2 concentrations in the Baltic Sea, based on long-term observations at the Finnish Utö Island." by Maragkidou et al., EGUSPHERE2024-1703.**

*For clarity, our response to the reviewer's comment is in **purple** font.*

Dear Authors,

many thanks for submitting your revised manuscript along with your responses to the queries from the reviewers. I am happy with the changes that have been made to the manuscript in the most part. However, I do not feel that the new sub-section 4.4 adds any real scientific value to the manuscript and I suggest removing or moving the text from this section earlier in the manuscript - it would fit before lines 144 - 157. I assume that when the reviewer 2 queried the uncertainties associated with the measurements, they wanted details on the calibration uncertainties in terms of x % at 1 or 2 sigma. I think these details on the measurement uncertainties should be added to the manuscript somewhere around lines 144 - 157.

A couple of minor corrections:

Line 154: define GPT

Line 193: Lee et al

Once these revisions have been made, I am happy to recommend publication in ACP.

Best wishes.

**Reply:** We would like to thank the editor for her comments and for recommending our manuscript for publication in ACP once her suggested revisions are made. We have added a new column to Table 1, titled "Uncertainty," which includes the uncertainties of the instruments used or of equivalent models, based on their datasheets, manuals, or manufacturer-provided specifications available online. Regarding the suggestion to remove or move subsection 4.4 earlier in the manuscript, we decided to remove this text from the manuscript. Additionally, we have addressed the editor's minor corrections by providing the definition of GPT (gas phase titration) in new line 156 and correcting the reference to Lee et al. 2011 in line 195.

**List of all relevant changes made in the manuscript:**

| Changes | Date |
|---|---|
| Added a new column to Table 1, titled "Uncertainty," which includes the uncertainties of the instruments used or of equivalent models, based on their datasheets, manuals, or manufacturer–provided specifications available online. | 12 December 2024 |
| Provided the definition of GPT (gas phase titration) in new line 156 (old line 154). | 12 December 2024 |
| Corrected the reference from Lee at al. (2011) in new line 195 (old line 193). | 12 December 2024 |
| Deleted subsection "4.4 Uncertainties" as suggested by the editor. | 12 December 2024 |

| | |
|---|---|
| Added two clarifications below Table 1 (new lines 143–144). | 13 December 2024 |
| Added one reference to the References section (new lines 593–595) | 13 December 2024 |
| Added a small sentence to the Acknowledgements section (new line 431) | 13 December 2024 |
| Removed extra space in new lines 27, 34 and 41 | 13 December 2024 |
| Added a period between "2" and "5" in "PM25" in the first column of Table 1 | 13 December 2024 |
| Replaced the hyphens (-) with en dashes (–) in new lines 400 and 419. | 13 December 2024 |